# Scoring-Based Genetic Algorithm for Wavefront Shaping to Optimize Multiple Objectives

**DOI:** 10.3390/jimaging9020049

**Published:** 2023-02-18

**Authors:** Tianhong Wang, Nazifa Rumman, Pascal Bassène, Moussa N’Gom

**Affiliations:** 1Department of Physics, Applied Physics, and Astronomy, Rensselaer Polytechnic Institute, 110 8th Street, Troy, New York, NY 12180, USA; 2Electrical, Computer, and Systems Engineering Department, Rensselaer Polytechnic Institute, 110 8th Street, Troy, New York, NY 12180, USA; 3Center for Ultrafast Optical Sciences, University of Michigan, Ann Arbor, MI 48109, USA

**Keywords:** wavefront shaping, light scattering, multiple foci, SBGA, NSGA2, imaging

## Abstract

We present a scoring-based genetic algorithm (SBGA) for wavefront shaping to optimize multiple objectives at a time. The algorithm is able to find one feasible solution despite having to optimize multiple objectives. We employ the algorithm to generate multiple focus points simultaneously and allocate their intensities as desired. We then introduce a third objective to confine light focusing only to desired targets and prevent irradiation in neighboring regions. Through simulations and experiments, we demonstrate the algorithm’s ease of implementation and flexibility to control the search direction. This algorithm can potentially be applied to improve biomedical imaging, optogenetics, and optical trapping.

## 1. Introduction

Light scattered by turbid materials retains substantial information, and wavefront shaping methods utilize this information from the seemingly random scattered photons to focus light beyond scattering media [1]. These techniques optimize the incident wavefront with a spatial light modulator (SLM) to compensate for multiple scattering events in the material. This has been done through methodologies based on phase conjugation [2,3], a transmission matrix [4,5], and a semi-definite programming [6] approach or feedback-based optimization [7,8,9].

The objective of most studies has been to find a single focus spot [10,11,12]. However, various biomedical applications, including photodynamic therapy (PDT) [13], optogenetics approaches [14], and optical trapping [15], often require irradiation of multiple cells, neurons, or targets at the same time. It is also important to prevent light exposure of healthy cells during PDT or undesired targets for optogenetics applications. In order to achieve light focusing at multiple spots, different metrics are utilized simultaneously for the optimization process. And as such, multiple point focusing has been treated as a multi-objective optimization problem [16]. One of the optimization techniques that is well suited to solve such problems is the genetic algorithm (GA) [17].

GA is a metaheuristic algorithm that mimics the natural evolution process and tries to find an optimal solution to a problem. Feedback-based wavefront shaping methods have already adopted the GA to find the optimal solution because of its better performance in noisy environments [18,19,20,21]. The GA’s ability to search for solutions with multiple objective functions or discriminants has made it an attractive proposition to tackle the problem of multi-point focusing.

One such GA-based algorithm is the non-dominated sorting GA II (NSGA2) [22], which has been used in wavefront shaping to achieve multi-point focusing [23]. NSGA2 looks for a set of solutions known as the Pareto set [24], which are optimal solutions in the space of objective functions in multi-objective optimization problems. The Pareto set therefore expands further with the increased number of objectives, and thus the optimization process becomes more complex. Furthermore, it was shown [25] that the search ability and performance of Pareto dominance-based algorithms such as NSGA2 degrades for problems with many objectives.

In this paper, we propose a simple scoring-based genetic algorithm (SBGA) that can reach a feasible solution despite having multiple objectives. It treats the problem of multi-point focusing as a single objective problem by combining the scores of each solution. Every solution is given a score according to its ability to optimize a particular objective function or discriminant. The search direction of the algorithm can be easily controlled. We demonstrate simultaneous focusing at multiple points where both enhancement and uniformity are taken into account with experiments and simulations. Additionally, we demonstrate that a predefined target intensity distribution over a region can be achieved by the algorithm. In addition to the problem of multi-point focusing, we consider reducing irradiation in the neighboring regions and introduce for the first time a third objective. We compare numerically and experimentally the performance of NSGA2 with that of SBGA when solving a test problem, where objectives are strongly correlated and conflicting. SBGA can be used in applications that require selectivity, where light is focused at specific targets and unwanted irradiation needs to be actively controlled.

## 2. Principle

SBGA is implemented for feedback-based wavefront shaping and is designed to optimize multiple objectives, e.g., creating multiple foci through a scattering medium simultaneously. For wavefront shaping, each generation consists of many individuals, where phase masks are treated as individuals. Each individual acquires a score based on its ability to achieve an objective. The algorithm continues to retain the better individuals and discard the worse ones through generations. The following three cases are considered to demonstrate the working principle of SBGA.

### 2.1. Multi-Point Focusing

Two objectives to be considered for multi-point focusing are enhancement and uniformity. We define two discriminants ($f_{1}$, $f_{2}$), similar to those in Hu et al. [26], to evaluate the performance of each phase mask:(1)$$f_{1}=\bar{I}=\frac{1}{M}\sum_{m=1}^{M}I_{m}$$
(2)$$f_{2}=\sigma_{M}/\bar{I}$$

Here, *M* is the number of targets or focus points, $f_{1}$ is the average intensity of *M* points, $I_{m}$ is the intensity at the mth focus, and $\sigma_{M}$ is the standard deviation of the intensity of *M* focus points. Higher average intensity corresponds to higher enhancement. Therefore, higher scores denoted as $S_{1}$ are assigned to the masks that obtain higher $f_{1}$ values. Since the other objective is to equally distribute the intensity among all focus points, we want the relative standard deviation (or the coefficient of variance), $f_{2}$, to be minimized. Thus, better scores denoted as $S_{2}$ are assigned to the masks with smaller $f_{2}$ values.

To assign scores, the phase masks are first sorted in ascending or descending order determined by their performance in optimizing an objective function, $f_{i}$. The score ($S_{i}$), which is related to a discriminant ($f_{i}$), is then assigned to a phase mask based on its ranking. For instance, the phase mask with the best performance receives $S_{i}=n$, where *n* is the number of phase masks in one generation. The second-best phase mask receives $S_{i}=n-1$. Lastly, the phase mask with the worst performance receives $S_{i}=1$. The score-assigning scheme prevents solutions that are good at only optimizing one discriminant from being overestimated. For the remainder of the paper, the same score-assigning scheme is used for all discriminants.

The final score for each phase mask in one generation is calculated by the following equation:(3)$$S_{f}=aS_{1}+bS_{2}$$
in which *a* and *b* are two ranking coefficients to weight the scores. The ranking coefficients can be adjusted if one wishes to emphasize one objective over the other. Phase masks with higher scores are more likely to be picked as parents for the next generation.

### 2.2. Controlled Intensity Distribution

The second discriminant, Equation (Equation 2), is modified to realize a predefined intensity distribution.
(4)$$f_{2}^{\prime }=\sigma_{M}^{\prime }\left(I_{m}^{\prime }\right)$$
where $I_{m}^{\prime }=I_{m}/p_{m}$, $p_{m}$ is the desired relative intensity at the *m*th focus, and $\sigma_{M}^{\prime }$ is the standard deviation of $I_{m}^{\prime }$. $S_{1}$, $S_{2}^{\prime }$, and $S_{f}$ are assigned to individual phase mask through the same procedure as described in Section 2.1. Here, $S_{2}^{\prime }$ is the score assigned based on $f_{2}^{\prime }$.
(5)$$S_{f}=aS_{1}+bS_{2}^{\prime }$$

### 2.3. Intensity Minimization in Neighboring Region

When dealing with multi-point focusing, especially if the goal is to form a specific pattern, it is common to have concomitant focusing near the targets. Another discriminant characterizing the neighbor pixels is added to confine light focusing only to the desired pixels:(6)$$f_{3}=\sum_{j}I_{j}$$
where $I_{j}$ is the intensity of the *j*th neighbor pixel, and $f_{3}$ is the total intensity of the neighbor pixels. For this case, a smaller $f_{3}$ is preferred. A higher score $S_{3}$ is assigned to the phase mask, which results in a smaller $f_{3}$ value. The final score for each phase mask is now:(7)$$S_{f}=aS_{1}+bS_{2}+cS_{3}$$
where *c* is an additional ranking coefficient associated with the third score $S_{3}$.

### 2.4. Comparison with NSGA2

The standout feature of SBGA allows a multi-dimension problem to be converted into a one-dimension one in the score space, and the search direction can be controlled by changing the ranking coefficients to find one feasible solution.

An illustrative comparison of NSGA2 and SBGA is presented in Figure 1. NSGA2 searches for a front of solutions. It is often followed by a decision-making procedure to find a single optimal solution. By contrast, SBGA searches in one direction. By combining the scores, SBGA also limits the survival rate of solutions that are good at optimizing only one objective. The performance of these algorithms also depends on the correlation of the objective functions. For example, if the objectives are strongly correlated and conflicting at the same time, they might need to be ordered according to importance. This can be easily implemented with SBGA by choosing the appropriate ranking coefficients in Equation (Equation 7).

### 2.5. Detailed Steps

A flowchart of SBGA is presented in Figure 2. The detailed steps are as follows:**Step 1.** Set up the parameters for the experiment: the population size or the number of phase masks in each generation (*n*), dimension (number of superpixels) of a single phase mask (H×W), number of phase steps for every superpixel (ϕ), total number of iterations or generations ($iter_{total}$), current iteration count (iter=1), rate of mutation ($0\leq r_{mut}<1$).**Step 2.** Choose targets and discriminants according to the desired objectives. For the cases described in Section 2.2 and Section 2.3, desired intensity $p_{m}$ and the region of neighboring pixels need to be defined, respectively. Specify ranking coefficients (*a*, *b*, and *c*).**Step 3.** Generate *n* random phase masks that contain H×W superpixels in the first iteration. The phase of each superpixel is randomly selected from {1·2π/ϕ,2·2π/ϕ,⋯,ϕ·2π/ϕ}.**Step 4.** Display all phase masks in the current generation, and record the intensity of the area of interest.**Step 5.** Determine individual phase masks’ scores $S_{i}$ based on each objective function and calculate final score $S_{f}$.**Step 6.** Select parents based on scores and generate children (half of the population size, 1/2×n, Appendix A). The children will replace half of the phase masks with the lowest scores from the previous generation. (current iteration count, iter=iter+1)**Step 7.** Continue to step 8 if $iter==iter_{total}$. Otherwise, go back to step 4.**Step 8.** The mask with the best score from the last generation is chosen as the optimal solution.

## 3. Simulation Results

Simulation results are presented to show the performance of SBGA compared with NSGA2 [27]. Light waves through a scattering medium are modeled as plane waves propagating in different directions. A speckle field is created by the interference between these randomly generated plane waves [28]. To achieve a desired output pattern, SBGA (or NSGA2) is then applied to find the optimal phase mask represented by a H×W matrix (H=W=128). The simulation was run on an Intel i5-13600k CPU. For both NSGA2 and SBGA, it took about 1.5 min to finish a 1000-generation simulation.

The results of SBGA are presented in Figure 3 We use the ranking coefficients (a,b,c)=(2,1,0) (the neighbor intensity is not considered) and (a,b,c)=(2,1,1) (neighbor intensity is now considered). Figure 3a displays the target N pattern. The resultant speckle patterns obtained with SBGA after 1000 generations with and without neighboring intensity are shown in Figure 3b,c, respectively. The progress of the three objective functions $\bar{I},\sigma_{M}/\bar{I}$, and $I_{NBR}$ over generations are shown in Figure 3d–f, respectively. These curves are the average results of 10 distinctive simulations while the speckle intensity images were obtained from one of these trials. It can be seen in Figure 3f that the neighboring intensity has increased through generations due to the optimization of nearby targets. However, the introduction of the third objective minimizes $I_{NBR}$ when compared to the scenario of not utilizing this discriminant.

The trend of search quality and convergence progress of NSGA2 and SBGA over generations is presented in Figure 4. Three discriminant functions ($f_{1}$, $f_{3}$, $f_{3}$) with two different sets of coefficients, (a,b,c)=(1,1,1) or (2,1,1) are considered and the population from generations 1, 500, and 1000 are shown to demonstrate the algorithms’ performance.

NSGA2 and SBGA with (a,b,c)=(1,1,1) would not effectively work in case the discriminants have a strong correlation but conflicting goals, i.e., maximal average intensity and minimal neighboring intensity (Figure 4a,b). The strength of the correlation between intensities, which is related to the speckles’ grain size, becomes crucial for experiments that might lead to the failure of certain algorithms. While the correlation effect can be limited by choosing a different neighboring region, e.g., a few pixels away from the focus points, the coefficients used in SBGA can be modified to ensure the success of the algorithm (Figure 4c). We will demonstrate and further discuss this case in the experimental results.

The Pearson correlation coefficient (PCC, denoted by γ) is used to evaluate the quality of the focus pattern [29]. It is calculated by:(8)$$\gamma =\frac{\sum_{k}\left[I_{k}^{\left(C\right)}-<I^{\left(C\right)}>\right]\left[I_{k}^{\left(T\right)}-<I^{\left(T\right)}>\right]}{{\left\{\sum_{k}{\left[I_{k}^{\left(C\right)}-<I^{\left(C\right)}>\right]}^{2}\sum_{k}{\left[I_{k}^{\left(T\right)}-<I^{\left(T\right)}>\right]}^{2}\right\}}^{1/2}}$$

Here, $I_{k}^{\left(C\right)}$ is the intensity at the *k*th pixel of the camera, and $I_{k}^{\left(T\right)}$ is the target intensity at the *k*th pixel. $<I^{\left(C\right)}>$ and $<I^{\left(T\right)}>$ are the average of $I_{k}^{\left(C\right)}$ and $I_{k}^{\left(T\right)}$, respectively. The summation ($\sum_{k}$) is calculated over the whole area of interest. In our case, Figure 3b is our area of interest.

Figure 5 shows the PCCs and average intensities $\bar{I}$ for different sets of ranking coefficients (a,b,c) after 1000 generations. Here, the highest possible PCC value is limited by the speckles’ grain size. A trade-off between the average intensity and the PCC can be seen after introducing the neighboring intensity, which can be balanced by the coefficients.

## 4. Experimental Setup

The experimental setup is illustrated in Figure 6. The light source is a continuous wave (CW) helium-neon laser (HeNe, λ=632.8nm, JDS Uniphase 1101/P, 1.65 mW, polarized, JDS Uniphase Corporation, Milpitas, California, USA). The beam is expanded and incident on a phase-only SLM (Santec SLM-200, 1920 × 1080 pixels, pixel size 7.8 μm × 7.8 μm, Santec Corporation, Komaki, Japan). The polarization of the incident beam is controlled by a half-wave plate (HWP). The reflected beam is transmitted through a 4*f* imaging system (not shown) and focused on a scattering sample (ground glass) by a microscope objective (MO1, 10×, NA = 0.3, Olympus SPlan10, Olympus Corporation, Tokyo, Japan).

A second microscopic objective (MO2, 20×, NA = 0.4, Olympus NeoSPlan20, Olympus Corporation, Tokyo, Japan) was used to collect the speckle from the sample. An sCMOS camera (Thorlabs CS2100M, 1920 × 1080 pixels, pixel size 5.04 μm × 5.04 μm, Thorlabs, Newton, New Jersey, USA) was used to record the speckle. Both the SLM and the camera were controlled by MATLAB scripts.

## 5. Experimental Results and Discussion

### 5.1. Multi-Point Focusing

To test the efficacy of the SBGA algorithm, we generated multiple focal points simultaneously through the scattering medium.

After running through 600 generations, the algorithm was able to optimize 98 targets at the same time. A handwritten ‘N’ can be clearly seen in the speckle image shown in Figure 7c. For this experiment, only discriminants $f_{1}$ and $f_{2}$ were used with the ranking coefficients (a,b)=(2,1) (Equation (Equation 3)).

Figure 7a,b show that the algorithm achieved both objectives, i.e., increasing the average intensity and decreasing the relative standard deviation at the same time. Other parameters chosen for the experiment are H=W=150, ϕ=20, n=50, and $r_{mut}=4.4\%$.

### 5.2. Controlled Intensity Distribution

To demonstrate control of the intensity distribution, we targeted a two-dimensional Gaussian intensity distribution within a 20×20 pixels area. A total of 400 pixels were taken into account as targets by the algorithm.

The increment of average intensity is shown in Figure 8a. Figure 8b shows that the final intensity distribution matches well with the target intensity distribution. In Figure 8c, the region focus we obtained in the speckle after 500 generations is displayed. This time, discriminants ($f_{1}$, $f_{2}^{\prime }$) were used with ranking coefficients (a,b)=(2,1) (Equation (Equation 5)). Other parameters were H=W=40, ϕ=20, n=50, and $r_{mut}=1.25\%$.

### 5.3. Intensity Minimization in Neighboring Region

Here, we introduce the third discriminant ($f_{3}$) relevant to the neighbor intensity (Equation (Equation 7)) and create an ‘N’ pattern again. The ranking coefficients are (a,b,c)=(2,1,0.5). The other parameters are H=W=150, ϕ=20, n=50, and $r_{mut}=4.4\%$.

An area consisting of 50×50 pixels around the targets is considered as the neighboring region. Figure 9 shows a comparison between the target N, and the optimized speckle image without and with the neighboring intensity minimized. A few unwanted focus points in the neighboring region (marked by red circles) are observed in Figure 9b, which are minimized in Figure 9c.

Therefore, adding the third discriminant results in better localization of targets in comparison with the one achieved without considering the neighboring intensity. Additional results of the different patterns “R”, “P”, and “I” are shown in Figure 10. Three discriminants with ranking coefficients (a,b,c)=(2,1,0.5) were considered for this experiment. Since the pattern “I” has a smaller number of focus points (61 pixels), its average intensity is higher than “R” (96 pixels) and “P” (82 pixels).

For the purpose of comparison, we have employed both NSGA2 and SBGA with two discriminants ($f_{1}$, $f_{2}$). Figure 11a,b show experimental results of the population from three different generations (1st, 300th, and 600th) when using NSGA2 and SBGA, respectively. It can be noticed that SBGA is able to find better solutions in comparison to NSGA2 with the same number of generations. Figure 11c,d show that the neighboring intensity increases with the average intensity in a similar trend for both algorithms when using two discriminants ($f_{1}$, $f_{2}$).

Figure 12a shows the experimental results of using NSGA2 to optimize three discriminants. The algorithm did not find a good solution due to the strong correlation between the neighboring and the average intensity of targets. The SBGA with coefficients (a,b,c)=(1,1,1) had the same issue (Figure 12b). The problem can be solved with SBGA by simply changing the ranking coefficients to emphasize one objective more than the others.

To better illustrate the flexibility to control search direction and quality offered by SBGA, the populations from three generations with different coefficients are shown in Figure 13a–c. The relation between the neighboring intensity and the average intensity of targets is shown in Figure 13d–f. The solid line in each 2-D plot indicates the trend of the corresponding 3-D plot (Figure 13a–c) and the dashed lines are used for comparison with the trends of the two other 3-D plots.

Figure 13d–f show that the addition of the third discriminant and modification of ranking coefficients ensure an increase in target intensity while limiting neighboring intensity.

The Pearson correlation coefficient (PCC), γ (Equation (Equation 8)), was used to evaluate the quality of the experimental result as well. Figure 14 shows the relationship between average intensity $\bar{I}$ and PCC for different sets of ranking coefficients. Similar to the simulation results (Figure 5), when *a* and *b* remain unchanged, a higher value of *c* results in a higher value of PCC with a compromise in $\bar{I}$. We have also tested the GA with PCC as the only discriminant as used in Wan et al. [29]. Although the highest PCC was achieved with that discriminant, the average intensity is very low (Figure 14). With our algorithm, it is possible to acquire better enhancement and high PCC values at the same time. The enhancement of the targets with this algorithm can be noted as $\eta ={\bar{I}}_{final}/{\bar{I}}_{initial}$ = 3.5∼6, varying with the choice of *c*. The total enhancement (η× number of foci) is, therefore, 350∼600 times, which is close to the existing algorithms for multi-point focusing [16,30]. Meanwhile, we were able to avoid the forming of unwanted focus in specific regions. The region can be the neighboring pixels to increase the image quality (as shown in Figure 9), or other areas where minimal exposure to scattered light is desired. This is important for applications such as optogenetics where specificity is a major issue.

The choice of ranking coefficients depends on the objectives and their order of importance. We have compared the results obtained with different coefficients. From our study, we have found that ranking coefficients between (a,b,c)=(2,1,0.5) and (a,b,c)=(2,1,1) give the best results. *a* was chosen to be 2 so that the average intensity $\bar{I}$ is weighted more to ensure bright foci. By comparing the results, we have found that c∈ [0.5, 1] reduces the light in neighboring pixels without affecting the average intensity. This range of *c* also leads to a good image quality (high PCC value).

## 6. Conclusions

In summary, we presented a scoring-based genetic algorithm (SBGA) that is able to find one optimal solution for multiple objectives. It is simple, flexible, and can be easily adjusted to change the search direction and emphasize specific objectives. Additionally, we experimentally demonstrated that SBGA can evolve faster when compared to NSGA2. We demonstrated the application of SBGA to three different cases. A clear pattern ‘N’ was generated by focusing light on 98 pixels simultaneously. We also realized region focusing with predefined Gaussian intensity distribution consisting of 400 pixels. In order to confine light focusing only to the desired region, we introduced a third discriminant. This prevents unwanted light from focusing on the adjacent area. SBGA can be applied to a variety of applications including optical trapping or fluorescence imaging.

## Figures and Tables

**Figure 1 jimaging-09-00049-f001:**
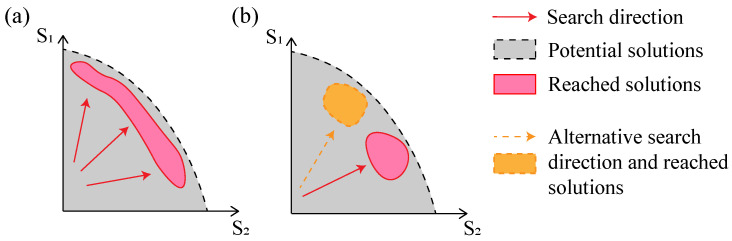
Searching scheme of NSGA2 and SBGA. (**a**) NSGA2 retains a front formed by the solutions in each generation. (**b**) SBGA reduces the problem to one dimension in the score space and searches in a specific direction (NSGA2: Non-dominated Sorting Genetic Algorithm II, SBGA:Scoring-Based Genetic Algorithm).

**Figure 2 jimaging-09-00049-f002:**
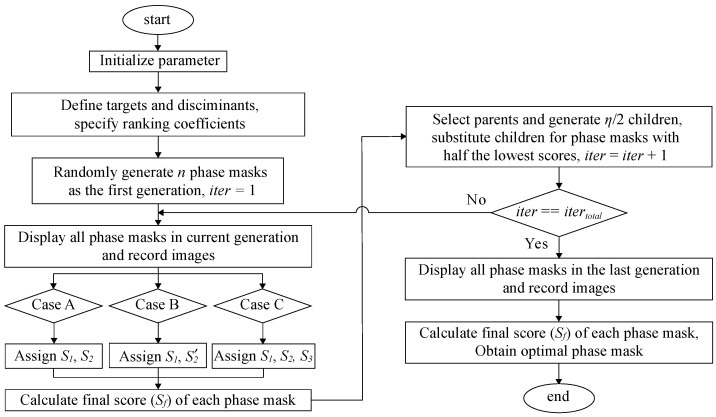
Flowchart of SBGA for wavefront shaping.

**Figure 3 jimaging-09-00049-f003:**
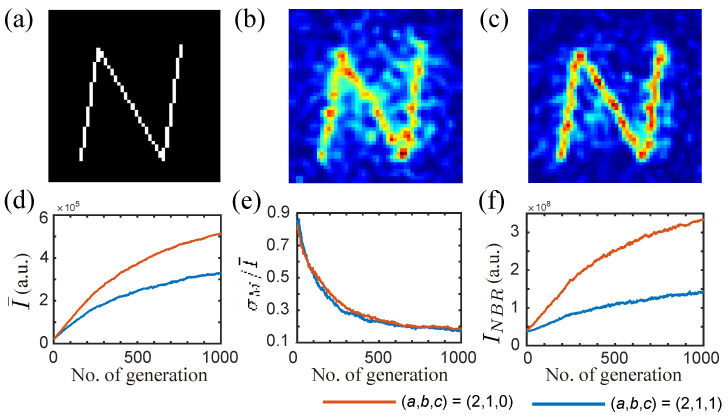
Simulation results of SBGA. (**a**) Target N pattern. (**b**) Optimal speckle image with two discriminants (a,b,c)=(2,1,0). (**c**) Optimal speckle image with three discriminants (a,b,c)=(2,1,1). Progress of (**d**) average intensity, (**e**) relative standard deviation, and (**f**) neighboring intensity with the number of generations. (**b**,**c**) were normalized separately.

**Figure 4 jimaging-09-00049-f004:**
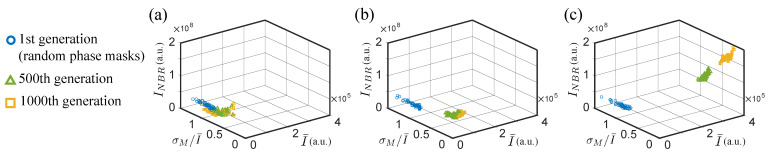
Simulation results considering 3 discriminants for (**a**) NSGA2, (**b**) SBGA with (a,b,c)=(1,1,1), and (**c**) SBGA with (a,b,c)=(2,1,1). The solutions from generations 1 (random phase masks), 500, and 1000 (last generation) are shown.

**Figure 5 jimaging-09-00049-f005:**
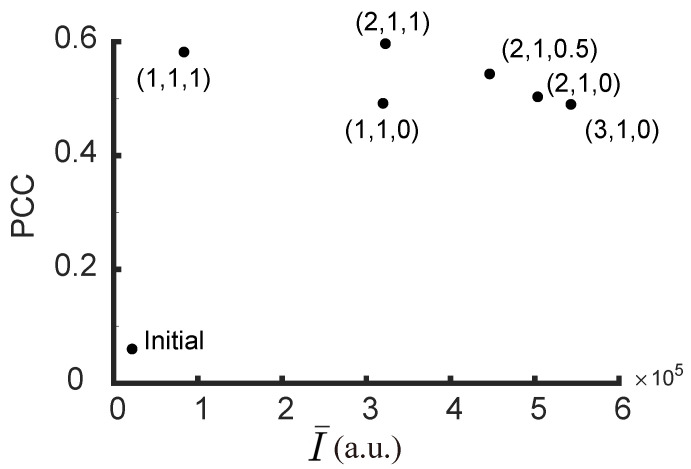
Pearson correlation coefficients (PCC) and average intensities $\bar{I}$ for different sets of ranking coefficients (a,b,c) after 1000 generations. The ranking coefficients are labeled near the points. The initial point is also shown in the bottom left corner.

**Figure 6 jimaging-09-00049-f006:**
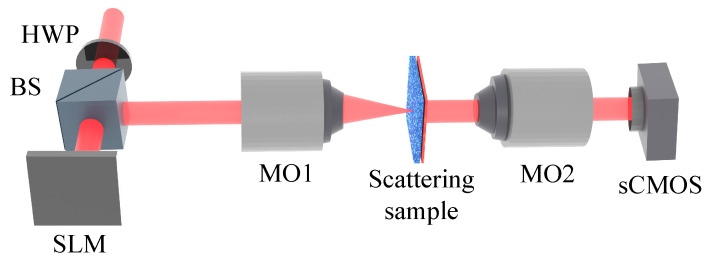
Schematic of the experimental setup. The expanded incident beam (λ=632.8 nm) passes through a half-wave plate (HWP) and a beam splitter (BS) to the spatial light modulator (SLM). From the SLM, the beam is focused onto the scattering sample by a microscope objective (MO1). The speckle is collected by MO2 and imaged by an sCMOS camera.

**Figure 7 jimaging-09-00049-f007:**
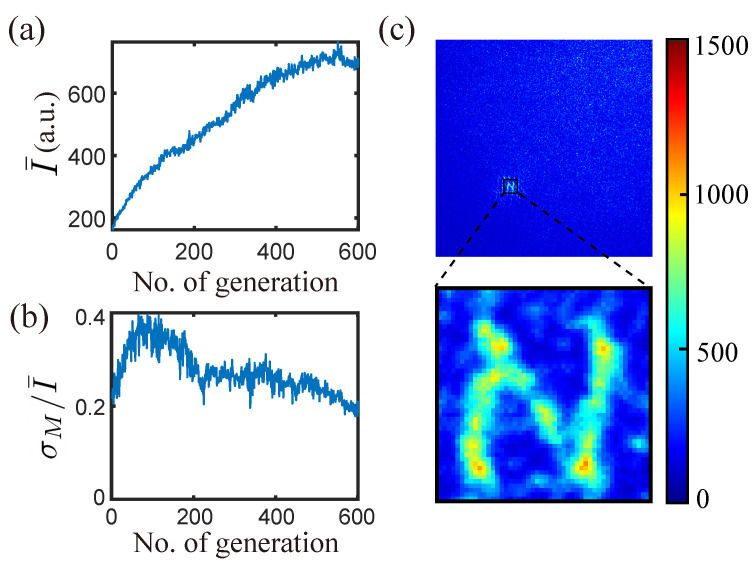
Multi-point focusing optimization using SBGA with two discriminants ($f_{1}$, $f_{2}$). (**a**) The average intensity of the predefined targets over generations. (**b**) The relative standard deviation of the intensity of all the targets. (**c**) Optimal speckle image.

**Figure 8 jimaging-09-00049-f008:**
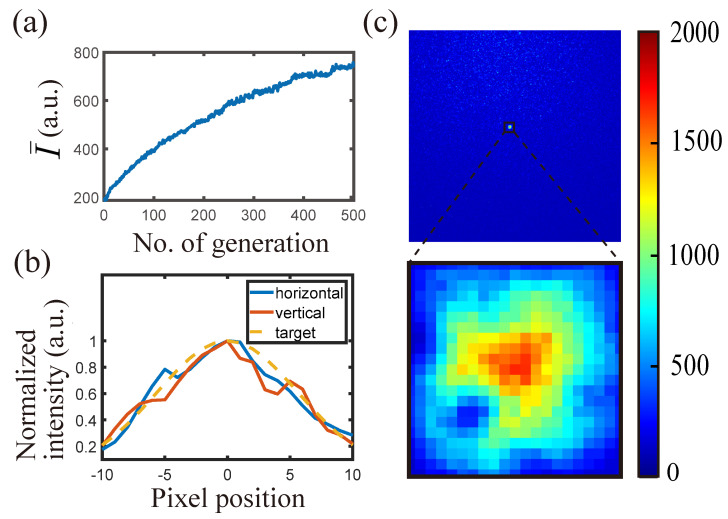
Intensity distribution controlled using SBGA with two discriminants ($f_{1}$, $f_{2}^{\prime }$). (**a**) Average intensity of the 20×20 pixels area over 500 generations. (**b**) Normalized optimal intensity along the horizontal and vertical lines across the center. The desired target intensity distribution is shown by the yellow dashed line. (**c**) Optimal speckle image.

**Figure 9 jimaging-09-00049-f009:**
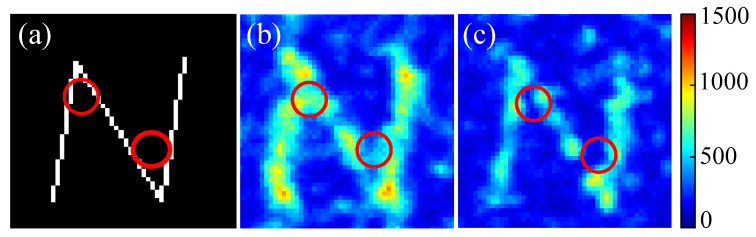
Neighboring intensity minimization using SBGA. (**a**) The target pattern, ‘N’. (**b**) Optimal speckle image without minimizing neighboring intensity ($f_{1}$, $f_{2}$). (**c**) Optimal speckle image with the third discriminant to minimize neighboring intensity ($f_{1}$, $f_{2}$, $f_{3}$). The main differences are marked by the red circles.

**Figure 10 jimaging-09-00049-f010:**
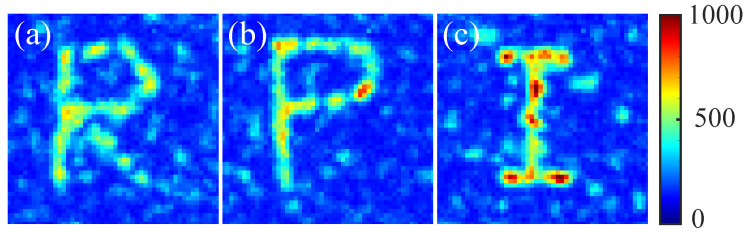
Patterns “R” (**a**), “P” (**b**), and “I” (**c**) were created experimentally by utilizing SBGA with three discriminants ($f_{1},f_{2},f_{3}$) and ranking coefficient (a,b,c)=(2,1,0.5).

**Figure 11 jimaging-09-00049-f011:**
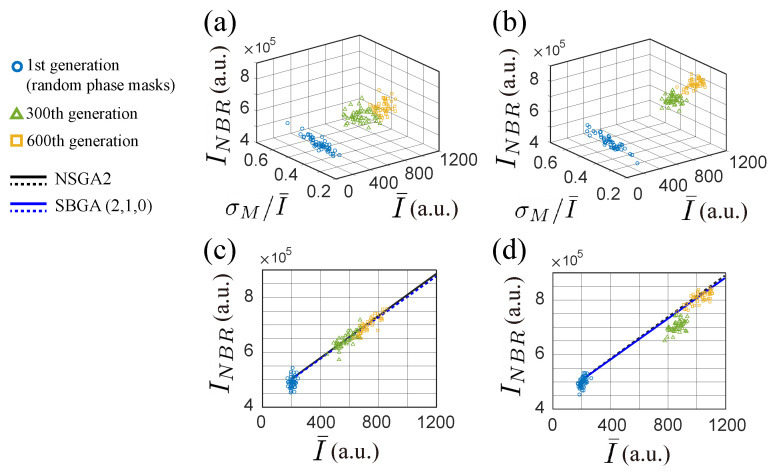
Experimental population evolution for NSGA2 (first column) and SBGA (second column). 3-D plots of population evolution with 2 discriminants ($f_{1},f_{2}$) for (**a**) NSGA2, and for (**b**) SBGA with ranking coefficients (a,b,c)=(2,1,0). 2-D plots shown in (**c**,**d**) present the relation between average intensity $\bar{I}$ and neighboring intensity $I_{NBR}$. In each 2-D plot, the solid line follows the average evolution trend and the dashed line represents a comparison between NSGA2 (black) and SBGA (blue).

**Figure 12 jimaging-09-00049-f012:**
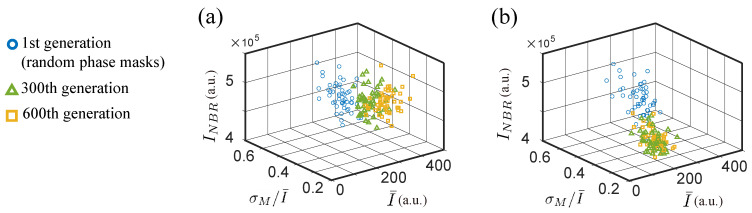
Experimental proof of the limits of NSGA2 and SBGA. Population evolution with three discriminants for (**a**) NSGA2 and (**b**) SBGA with equally weighted ranking coefficients (a,b,c)=(1,1,1). Solutions from the 1st, 300th, and 600th generations are shown.

**Figure 13 jimaging-09-00049-f013:**
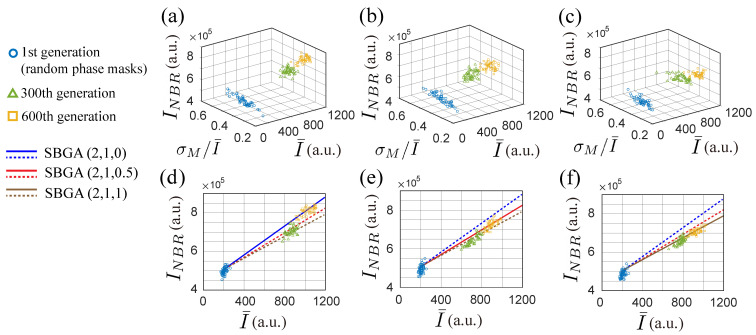
3-D and 2-D plots of experimental population evolution of SBGA with three discriminants ($f_{1}$, $f_{2}$, $f_{3}$) and three different sets of ranking coefficients (**a**) (a,b,c)=(2,1,0), (**b**) (a,b,c)=(2,1,0.5), and (**c**) (a,b,c)=(2,1,1). The 2-D plots shown in (**d**–**f**) present the relation between average intensity $\bar{I}$ and neighboring intensity $I_{NBR}$. In each 2-D plot, the solid line follows the average evolution trend and the dashed lines represent a comparison between different sets of ranking coefficients.

**Figure 14 jimaging-09-00049-f014:**
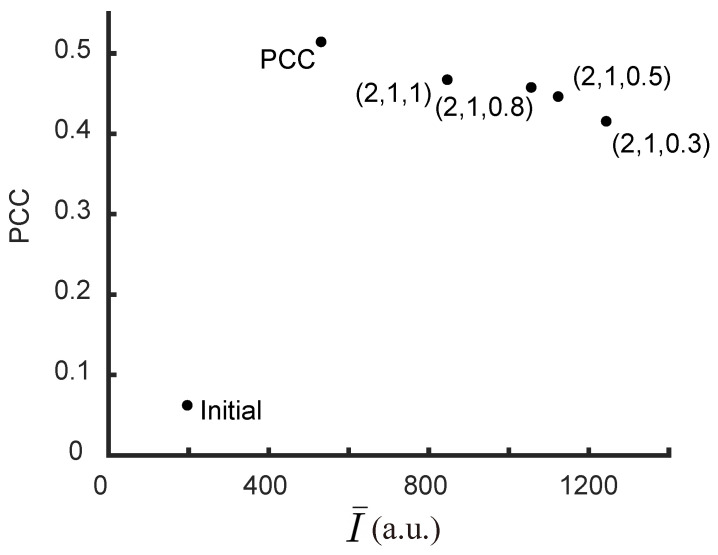
Pearson correlation coefficients (PCCs) and average intensities $\bar{I}$ for different sets of ranking coefficients (a,b,c) after the 600th generation. The ranking coefficients are labeled near the points. The initial point is also shown in the bottom left corner.

## Data Availability

The data presented in this study are available upon request from the corresponding author.

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
