# Peer review of "Scoring-Based Genetic Algorithm for Wavefront Shaping to Optimize Multiple Objectives"

_2313-433X, 2023, doi:10.3390/jimaging9020049_

Round 1

Reviewer 1 Report

The authors present a new scoring system for genetic algorithms to use in feedback-assisted wavefront shaping of multiple targets. While the study is interesting and the results could be impactful the manuscript needs significant revisions. My comments are below:

1) Additional citations are needed in the introduction to provide more context with regards to the literature on optimization algorithms in this field. Here is a non exhaustive list of some papers of interest:

I. Vellekoop and A. Mosk, “Phase control algorithms for focusing light through turbid media,” Opt. Commun. 281, 3071–3080 (2008).

-B.R. Anderson et al., “Microgenetic optimization algorithm for optimal wavefront shaping,” Applied Optics 54, 1485 (2015)

-Z. Fayyaz et al., “Simulated annealing optimization in wavefront shaping controlled transmission,” Applied Optics 57, 6233 (2018)

- D. Wu et al., “A thorough study on genetic algorithms in feedback-based wavefront shaping,” Journal of Innovative Optical Health Sciences 12, 1942004 (2019).

- “A comparative study of optimization algorithms for wavefront shaping,” Journal of Innovative Optical Health Sciences 12, 1942002 (2019).

-Y.Wu et al “Focusing light through scattering media using the harmony search algorithm for phase optimization of wavefront shaping,” Optik 158, 558 (2018)

- “Focusing light through random scattering media by four-element division algorithm,” Opt. Comm. 407, 301 (2018)

-B.R. Anderson et al. “Genetic algorithms for focusing inside opaque media” Journal of Optics 22, 085601 (2020)

-B. Zhang et al., “Focusing light through strongly scattering media using genetic algorithm with SBR discriminant” J. Opt. 20 025601 (2018)

-X Zhang et al “Binary wavefront optimization using a genetic algorithm” J. Opt. 16 125704 (2014)

-R. Li et al., “Interleaved segment correction achieves higher improvement factors in using genetic algorithm to optimize light focusing through scattering media” J. Opt. 19 105602 (2017)

-L. Fang et al., “Binary wavefront optimization using a simulated annealing algorithm” Appl. Opt. 57 1744–51 (2018)

-Z. Fayyaz, “A comparative study of optimization algorithms for wavefront shaping J. Innovative Opt. Health Sci. 12 1942002” (2019)

-L. Fang et al., “Binary wavefront optimization using particle swarm algorithm” Laser Phys. 28 076204 (2018)

-L. Fang et al.,  “Focusing light through random scattering media four-element division

algorithm” Opt. Commun. 407 301–10 (2018)

2) Some of the nomenclature and notation the authors use is not consistent with the literature. For instance, Eta is traditionally used to denote the intensity enhancement in this field. It should not be used to denote the number of masks in a generation.

3) Some further discussion is needed of the ranking choices. In other systems the weight of ranking is proportional to the resulting fitness of a population member (i.e. if a population member gets 100x fitness it gets 100x weight). In this system the lower fitness members could end up having proportionally too much weight (i.e. if the the middle population member is 100x less fit than the best member, it will end up with half the weight of the best member).

5) In the algorithm steps, step 6 only talks about 1/2 of the children. It is not clear what they do with the other half. I assume that the top half carries over to the new generation, but this should be explicitly stated.

6) In figure 3 you show the optimization results when you include the “minimize neighbor” objective. It would be nice to compare this to the optimization without this objective to see how the introduction of this new objective affects the results.

7) In Figure 5 the correlation coefficients are all <0.6, which seems low. In my experience, papers using Pearsons correlation coefficient typically talk about good correlation for coefficients >0.9, so 0.6 seems pretty poor. You should provide more discussion and context for this figure and explain why the values are relatively low.

8) The experimental setup needs significantly more details. Ideally list the objective manufacturers and models, the cameras pixel format and size, the SLMs pixel format and size, the lasers manufacturer, power, beam size, etc. Also what is the thickness and mean free path of the ground glass?

9)  In general the authors keep talking about how their scoring system is better than the NSGA2 approach, but don’t provide any proof of this as they don’t show comparison results for NSGA2 experiments. 

10) The manuscript has numerous typos that need to be fixed.

11)  Why does section 5.1 and 5.3 have the same mutation rate, but section 5.2 has a different rate?

12) Figures 4, 10, and 11 are hard to interpret and don’t do a good job of showing “population evolution”. Plots showing the different fitness functions as a function of generation are much clearer and more useful. Data at 1 generation, 500 generations, and 1000 generations really doesn’t tell me about the evolution of the fitness functions with generation. These figures need to be rethought and modified.

13) You claim that the “SBGA can evolve faster when compared to NSGA2” but your paper has not demonstrated this. There needs to be a figure showing the evolution as a function of generation for both algorithms to demonstrate this. Currently this claim is totally baseless in the manuscript.

Reviewer 2 Report

This paper reports on the coherent control of a laser beam wavefront through a scattering medium using a genetic algorithm. The authors highlight on the efficiency of the proposed method to optimize an intensity pattern made of several focusing dots instead of a simple single spot. Three objectives are taking into account for that purpose. The authors compare the performance of their algorithm with another version of genetic algorithm previously published. This is an incremental work about laser wavefront shaping with an optimization process.

The paper is clearly written, and shows relevant results.

It would be interesting to have a more general idea of the benefit of using the proposed process over others.

-          Maybe a statistical study on different shapes would improve the paper.

-          Moreover, it is not clear how the ranking coefficients are chosen. Are they dependent of the shape?

-          What about the number of focusing dots? Is it possible to investigate the efficiency of the process as function of the number of focusing dots? Is it a relevant parameter?

-           

This paper is of interest to people of the domain, but it requires some additional details to be published in Journal of Imaging.

Reviewer 3 Report

In the manuscript entitled “Scoring based genetic algorithm for wavefront shaping to optimize multiple objectives”, Wang et al. proposed a scoring-based genetic algorithm (SBGA) for multi-objective optimization problems in wavefront shaping. By introducing different discriminants, selecting these discriminants according to different optimization objectives, and scoring individuals in each generation, a better control of the search direction in the optimization process is achieved. The comments are given as below.

1.       The manuscript discussed about different ranking coefficients. Basically, they are compared with each other mainly in Simulation Results and Intensity minimization in neighboring region parts. It can be seen that the ranking coefficients is crucial for controlling the search direction and optimizing the results. However, with many comments about different ranking coefficients, there is no clear final solution given in the manuscript.

2.       There is no systematic description of the mechanism (how the coefficients affect the results), selection range and robustness, which would help to select the appropriate coefficients.

3.       For the test pattern, only one pattern is used in the manuscript to show the experimental results. In order to better explain the feasibility of SBGA and its advantages over NSGA2, the additional patterns should be added to show the comparison results between NSGA2 and SBGA.

4.       The optimizing time of SBGA should be given and compared with that of NSGA2.

5.       The manuscript shows the performance of SBGA for multi-objective optimization, but there are few descriptions about its application, especially the expected application prospects corresponding to the advantages of other multi-objective genetic algorithms.

6.       The method proposed in the paper (SBGA) is superior to the traditional scheme (NSGA2), but the article only has some theoretical analysis. Please add comparison of experimental results.

7.       Many figures in the manuscript are similar and no special highlights. The readability of the article graph is poor.

8.     The manuscript needs carefully checked for grammatical errors and inconsistencies.

Round 2

Reviewer 1 Report

The authors have done a good job addressing most of my criticisms and the paper is much improved. However, I still find Fig 4, 11, and 12 to be confusing. Plotting dots vs the different discriminates for 3 different numbers of generations is just confusing. I still think much more appropriate figures would be graphs that have the number of generations as the independent variable. Things would make more sense with the discriminates as dependent variables plotted against the number of generations as the independent variable. In almost every other algorithm paper that is the norm. Their plotting approach is atypical and not clear.

In my opinion, choosing to plot in this weird way drastically decreases the impact and readability of the paper. Changing the figures to have the number of generations as the independent axis would make a huge improvement.

Reviewer 3 Report

In the manuscript entitled “Scoring based genetic algorithm for wavefront shaping to optimize multiple objectives”, Wang et al. made some supplements and modifications. In general, the choice of the best coefficients should be analyzed and described more clearly. Regarding the author's response, the following issues need to be considered:

1.         The manuscript describes the generation process of discriminants and scores in the Principle part. However, it seems that the scores of phase masks can be directly weighted according to the discriminants and ranking coefficients, without having to be sorted to get the scores. Please consider the possibility of such simplification.

2.         In the simulation and experimental results, in addition to the discriminants, the manuscript only uses the Pearson correlation coeffcient (PCC) metric to compare the solutions, but the ranking coefficients still needs to make a trade-off between the PCC and the average intensity, that is to say, the PCC index cannot directly indicate the best coefficients. The authors are suggested to consider whether there are other metrics to evaluate the quality of focused patterns.

3.         As mentioned in 2, the PCC metric cannot directly indicate the pros and cons of the coefficients, and there are trade-offs among the discriminants. Please discuss the metrics and considerations used to select the best ranking coefficients.

4.         The experimental results for different patterns are supplemented in Figure 10. It can be seen that when the coefficients (a, b, c) = (2, 1 ,0.5), the quality of different pattern optimization results has differences. Do the optimization solutions for these patterns correspond to best coefficient choices? If not, it would be more reasonable to replace it with the optimization solutions corresponding to the best coefficients. If so, it can be seen that the optimization solution of the pattern "R" is slightly inferior to that of the pattern "I". Please discuss and analyze about this point.
